# Unveiling and Mitigating Bias in Audio Visual Segmentation

## ABSTRACT

Community researchers have developed a range of advanced audio-visual segmentation models aimed at improving the quality of sounding objects' masks. While masks created by these models may initially appear plausible, they occasionally exhibit anomalies with incorrect grounding logic. We attribute this to real-world inherent preferences and distributions as a simpler signal for learning than the complex audio-visual grounding, which leads to the disregard of important modality information. Generally, the anomalous phenomena are often complex and cannot be directly observed systematically. In this study, we made a pioneering effort with the proper synthetic data to categorize and analyze phenomena as two types "audio priming bias" and "visual prior" according to the source of anomalies. For audio priming bias, to enhance audio sensitivity to different intensities and semantics, a perception module specifically for audio perceives the latent semantic information and incorporates information into a limited set of queries, namely active queries. Moreover, the interaction mechanism related to such active queries in the transformer decoder is customized to adapt to the need for interaction regulating among audio semantics. For visual prior, multiple contrastive training strategies are explored to optimize the model by incorporating a biased branch, without even changing the structure of the model. During experiments, observation demonstrates the presence and the impact that has been produced by the biases of the existing model. Finally, through experimental evaluation of AVS benchmarks, we demonstrate the effectiveness of our methods in handling both types of biases, achieving competitive performance across all three subsets.

## CCS CONCEPTS

• **Computing methodologies → Scene understanding**.

## KEYWORDS

Audio-Visual Segmentation, Multimodal Bias, Multimodal Learning

## 1 INTRODUCTION

In the real world, audio and visual are two closely related modalities that provide fundamental perception in regular life, often applied in forms such as videos. One classic task within the field of audio-visual understanding is Audio-Visual Localization (AVL) [17, 36], which enables the unsupervised localization of sounding objects by utilizing sound as guidance. With the increasing demand for stronger perceptual capabilities in autonomous driving [37, 48], and

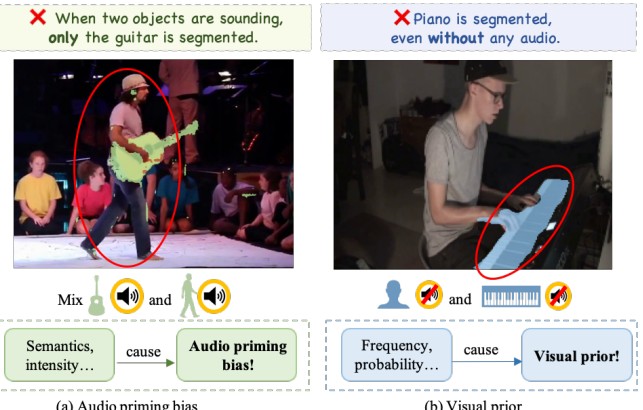

(a) Audio priming bias     (b) Visual prior

**Figure 1: The illogical anomalies caused by biases. (a) Even if each audio yields a satisfactory mask separately, the dominance still exists when they overlap. (b) In training data, the piano generally sounds. During testing, regardless of the sound presented, the model still tends to prioritize the piano. These two phenomena of the illogical anomalies can be categorized into "audio priming bias", and "visual prior", which typically are simultaneously observed as a general impediment in the AVS model.**

embodied intelligence [13, 31], there is an urgent to establish finer grounding beyond bounding boxes and heat maps between audio and visual elements. Therefore, a new task called Audio-Visual Segmentation (AVS) has emerged. It introduces a pixel-level, fine-grained scene understanding, bringing novel challenges to the field of audio-visual understanding. Furthermore, the performance of AVS showcases the current ability of machines to understand and integrate modalities in complex scenarios.

The existing works on AVS can be broadly categorized into fusion-based [18, 22, 25, 28, 46] and prompt-based methods [10, 29, 30, 38] according to Wang *et al.* [38]. The former primarily integrates audio and visual information at different stages to localize sounding objects, while the latter focuses on generating effective audio prompts and further finetuning the visual foundation model pre-trained on large segmentation datasets [20, 44]. Currently, prompt-based methods are the cutting-edge models in this field, demonstrating exceptional performance. However, researchers have observed illogical anomalies in these approaches [38, 43]. As illustrated in Fig. 1, even when the mask of the object seems plausible, two anomalous but interesting phenomena can be observed. 1) The segmentation results after overlaying audio do not align with the superimposition of masks guided by separate audios, even when each audio yields a satisfactory mask. Instead, dominance occurs. 2) Regardless of whether the piano produces sound, the model usually tends to segment the piano. Phenomena like Fig. 1 have a wide impact on the current model, which makes analyzing and addressing these phenomena an urgent work to improve the grounding behavior.

Generally, the phenomena depicted on the left and right sides of Fig. 1 are often coupled and cannot be directly observed individually. We manage to control the influencing factor with artificial data, including volume, etc., and reasonably categorize phenomena into two types of biases: "audio priming bias" and "visual prior". Specifically, **"audio priming bias"** refers to the phenomenon that the model tends to focus on audio salient content but not whole content. This bias is characterized by the model's insensitivity to audio of specific intensity or semantics and consequently the dominance of certain semantic audio in multi-source scenarios. As for **"visual prior"**, it refers to the phenomenon that the model may directly segment the common-sounding objects. The most common observed form of such bias is that regardless of the audio guidance employed, the model consistently segments the whole or part of certain objects. It is evident that both of these biases significantly impact the behavior of the current AVS models.

To provide the community with a comprehensive solution to address these biases, we adopt the divide-and-conquer thought. For audio priming bias, we first introduce **semantic-aware active queries**. These active queries contain rich latent semantic information gathered by the perception module, enhancing the sensitivity to audio cues. Furthermore, we employ a customized **interaction mechanism** in the transformer decoder specially designed to enhance the active queries and suppress the dormant ones for better collaboration of specific audio semantics. As for visual priors, we explore three types of **contrastive debias strategies** without changing the structure of the model. Our findings indicate that the soft and gradual debias strategy based on uncertainty yields superior results across various AVS methods including ours.

Finally, through experimental evaluation of AVS benchmarks, we demonstrate the effectiveness of our methods in handling both types of biases, achieving highly competitive performance across all three subsets. In summary, our contributions are threefold:

- We make a pioneering attempt to categorize the complex phenomena into two types of bias: "audio priming bias" and "visual prior" with necessary observation and analysis.
- For audio priming bias, we propose semantic-aware active queries and customized interaction mechanisms to improve the sensitivity and cooperation of complex audio scenarios. Then, for visual prior, the novel debias strategies are utilized to contrastively reorganize the distribution of logits.
- Experimental results reveal that the proposed method and strategy effectively mitigate biases while achieving versatility and comparable performance.

## 2 RELATED WORKS

### 2.1 Audio-Visual Segmentation

Before the emergence of AVS, traditional AVL tasks [17, 32, 36] relied on unsupervised learning that used bounding boxes or coarse heat maps for predicting the positions of sounding objects. However, for applications such as autonomous driving [37, 48], and embodied intelligence [13, 31], a more precisely multimodal grounding task to localize sounding objects at a pixel level is required. This is where AVS comes into the stage. The existing AVS approaches can be broadly categorized into fusion-based [18, 22, 25, 28, 43, 46] and prompt-based [10, 29, 38, 41]. The pioneering fusion-based

work [46] employs a multi-stage strategy to integrate audio with multi-scale visual features. Building upon this, CATR [22] further proposes a segmentation paradigm that combines both temporal and fusion information. Taking the thought of bidirectional generation, Hao *el al.* [14] achieve a further boosting on the performance. In contrast, prompt-based methods like AVSegFormer [10] and GAVS [38] directly decode fused features using projected audio queries. Notably, GAVS leverages rich visual information to achieve a generalizable model in zero-shot and few-shot scenarios. Moreover, taking a holistic perspective, Yan *et al.* [41] present a general framework for AVS and Ref-VOS [39] (Referring Video Object Segmentation), which both are cross-model guided video segmentation. However, previous AVS studies have primarily focused on achieving better multimodal fusion and plausible masks from the framework perspective, without delving into the challenges underlying the establishment of reasonable grounding.

### 2.2 Bias in Cross-Model Guided Tasks

Due to the widespread interest in the Visual Question Answering (VQA) task [3, 4], researchers have tackled two types of bias, namely "visual priming bias" and "language prior", as discussed in previous works [33, 42]. "Visual priming bias" was initially identified by Antol *et al.* [4], where models tend to prioritize visually salient content while disregarding the other context [12, 42]. Subsequently, the issue of "language prior" has gained attention from researchers [2, 33, 42], referring to the tendency of models to generate statistically common answers (e.g. the problem of green and yellow banana). The approaches for addressing the aforementioned bias can be broadly categorized into non-extra-based methods [5, 8, 33], and extra-based methods, involving additional human supervision [35, 40]. Different from extra-based methods, non-extra-based methods introduce a branch of the question-only model and conduct joint training with the target VQA model by specific learning strategies. Researchers have proposed several customized learning strategies including sharing the same question encoder[33], blocking the backpropagation of the question-only model [5], pretraining question-only model [8]. Moreover, it has been discovered recently that there is also a certain similarity between bias in VQA and hallucination [21, 47] in the popular Multimodal Large Language Model (MLLM).

However, the bias in AVS has never been explored by researchers before. Here, we hold the holistic opinion that VQA and AVS, as cross-model guided tasks, share certain similarities. VQA involves obtaining answers to questions based on image cues, while AVS involves obtaining masks from images based on audio cues. Therefore, in simple terms, VQA can be understood as a symmetrical problem to AVS in terms of modalities. So, the anomalous phenomena observed in AVS can potentially be explained by the bias theory present in VQA.

## 3 PROBLEM AND ANALYSIS

Based on the same thought of bias categories in VQA [42], we similarly classify the phenomena of anomalies into two types, "audio priming bias", and "visual prior" in Fig. 1. The segmentation failure cases observed in the previous model often tend to be a combination of both biases.

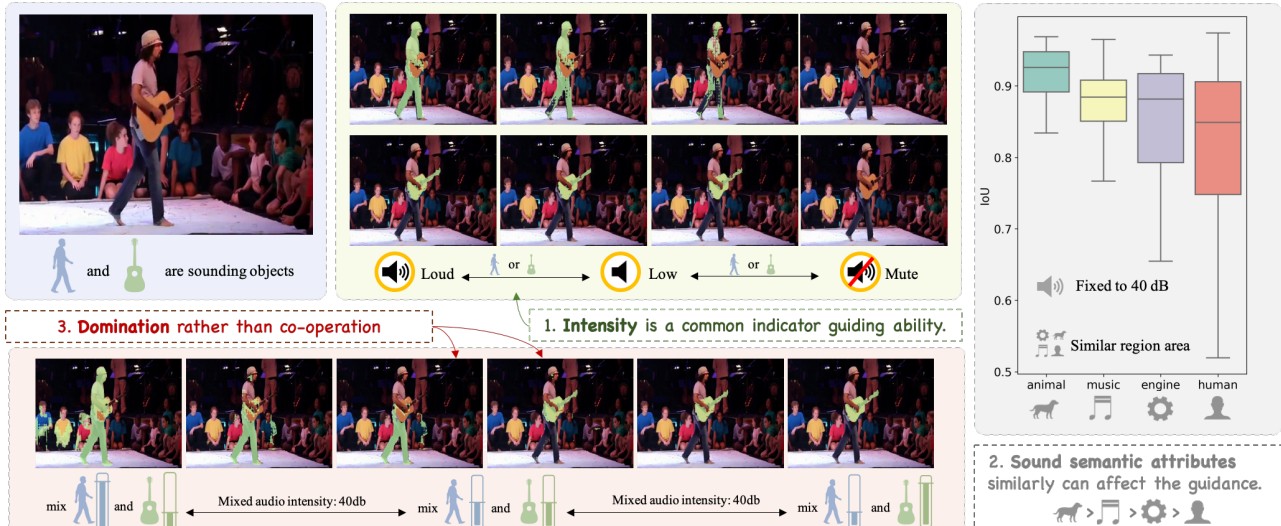

Figure 2: The illustration of audio priming bias. We have discovered that higher audio intensity results in stronger guiding capabilities in the green block. Also, the audio with distinct semantic attributes is easier to learn and possesses stronger guiding capabilities in the grey block. For instance, music has a greater guiding capability compared to human sound. Consequently, the diverse guiding capabilities cause the phenomenon of dominance in the red block.

## 3.1 Audio Priming Bias

The phenomenon that the model tends to focus on audio salient content but not whole content is called "audio priming bias". Firstly, through observation experiments involving manual intervention in audio (detailed in the appendix), we have observed the phenomenon of audio priming bias as illustrated in Fig. 2. Hence, the following observations can be discerned. 1) Audio with different intensities demonstrates varying guiding capability, as shown in Fig. 2 green block. 2) When controlling other variables including volume, we can observe a clear variance of the guiding capability by different semantics through the box plot. 3) In cases where multiple audios are simultaneously present, the overlaying of audio does not always lead to separate related masks being superimposed. Instead, the dominant of the audio appears as shown in Fig. 2 red block. Therefore, we contend that the primary determinants of audio guiding abilities are the **intensity** and **semantic attributes** of the audio. Consequently, the diverse guiding capabilities give rise to the phenomenon of dominance. Therefore, a well-designed mechanism is demanded to alleviate audio priming bias.

## 3.2 Visual Prior

The phenomenon that the model may directly segment the common-sounding objects is called "visual prior". Figure 4 reveals that, regardless of the audio provided, the AVS model consistently segments the partial or entire region of the particular object. Since the piano generally appears with a high sounding probability in the training data, the model tends to segment once sees the piano. Statistically, the **occurrence frequency** in [45] and the **sounding probability** in Fig. 4 between different semantics are imbalanced in the dataset. According to previous works [21, 23, 33], such preference and distribution provide strong prior information and make the model inclined to obtain statistically plausible results, rather than achieving the desired challenging grounding behavior. Moreover, due to the imbalance of inherent preference and distribution in the real world, addressing visual prior becomes more significant.

## 4 METHODS

The overall pipeline for the AVS task is normally constructed by the encoder-decoder structure in Fig. 3 involving an image encoder, a pixel decoder, and a transformer decoder. Generally, the multimodal information is fused and perceived in the transformer decoder [7, 10, 19, 22]. Therefore, the mitigation of **audio priming bias** requires an enhancing mechanism in the transformer decoder, while the mitigation of **visual prior** requires distribution reorganization after acquiring the logits. Methodologically, on one hand, to enhance the audio sensitivity and cooperation of different intensity and semantic attributes, we introduce the semantic-aware active queries utilizing a perception module and interaction enhancement mechanism. On the other hand, multiple training strategies are explored to contrastively optimize the debias model and reorganize the logits without modifying the structure.

## 4.1 Preliminaries

Given an audio-visual video sequence, we first divide it into $t$ non-overlapping audio and visual segment pairs with a length of 1s.

**Visual.** After the division, we extract visual representations $F_V \in \mathbb{R}^{t \times d_V \times H \times W}$ using a pre-trained Swin-base model [9]. The visual encoder is frozen and only tuned with two-layer MLP adapters.

**Audio.** Similarly, audio representations $F_A \in \mathbb{R}^{t \times d_A}$ is encoded from VGGish [11, 15], where $t$ represents the duration of the audio in seconds and matches the number of video frames. The audio representations are extracted in advance by the frozen encoder.

**Task.** Given a dataset $\mathcal{D} = \{F_V^i, F_A^i, \mathcal{M}_{gt}^i\}_{i=1}^n$ consisting of triplets of images $F_V^i \in F_V$, audio $F_A^i \in F_A$ and masks $\mathcal{M}_{gt}^i \in \mathcal{M}_{gt}$, the AVS task is to learn the mapping $\mathcal{G}(F_V, F_A) \to \mathcal{M}_{gt}$.

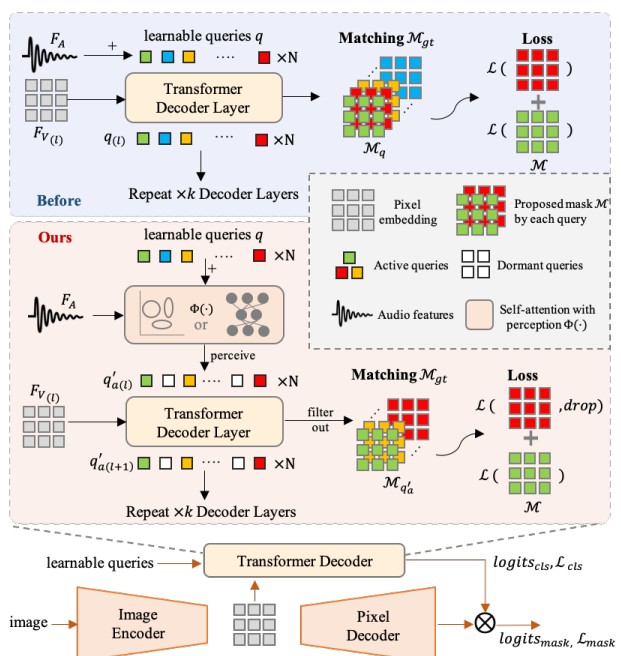

**Figure 3: In conventional semantic segmentation, the learnable queries are solely responsible for generating regions. However, in an ideal AVS model with a similar structure, the learnable queries need to not only generate regions but also perceive and regulate the interaction between audio semantics. So, this mechanism is designed to enhance the understanding of latent semantics and improve the interaction to only occur between semantic-aware active queries.**

## 4.2 Semantic-Aware Active Queries

In transformer decoder of conventional semantic segmentation, each learnable query ($\in \mathbb{R}^{1 \times d_A}$ among initialized learnable queries $q \in \mathbb{R}^{t \times N \times d_A}$, where $N$ is the number of learnable queries in $1s$ interval) functions like a region proposal network [34] and has the ability to generate mask proposals. These queries are simply inherited by repeatedly adding audio features $F_A$, in the AVS model [10, 22, 38] as shown in the upper block in Fig. 2. However, in an ideal AVS model, the learnable queries need to not only generate regions but also perceive and regulate the interactions between audio semantics adaptively.

To achieve the basic perception of audio semantics, we first introduce the following perception module aiming to enhance sensitivity to audio of various intensities and semantics. The logical and straightforward approach is to employ a separately trained projection $\Phi(\cdot)$ module with an optimization objective for audio alone to perform a coarse perception of latent audio semantics and to finally project to a latent semantic mark $\Phi(F_A) \in \mathbb{R}^{t \times N}$ with padding of 0.

$$\Phi(F_A) = \{1, x_1, x_2, ...\}, \tag{1}$$

$$x_i = \begin{cases} 1 & \text{if } F_A \in i^{th} \text{ class or cluster,} \\ 0 & \text{otherwise.} \end{cases} \tag{2}$$

Here, we use K-means, Gaussian Mixture Models (GMMs), or BEATs [6] to perform projection in both unsupervised and supervised ways. So each audio segment will be assigned with $1 \leq C \leq 8$ classes or $C = 1$ clusters. The overall goal of a latent semantic mark is a coarse filter to have a coarse and basic perception of the audio semantics. The items encoded by 1 in $\Phi(F_A)$ are termed *active* marks, whereas those encoded by 0 in $\Phi(F_A)$ are termed *dormant* marks. The active or dormant states are relative and vary from samples. Afterward, each audio segment will receive the latent semantic marks $\Phi(F_A) \in \mathbb{R}^{t \times N}$ based on the projection mentioned above. Then the semantic-enhanced audio features are obtained by element-wise product $\odot$ with broadcast

$$F'_A = \text{repeat}(F_A, N) \odot \Phi(F_A), \tag{3}$$

where $F'_A \in \mathbb{R}^{t \times N \times d_A}$ is semantic-enhanced audio features. Furthermore, we consistently encode the $0^{th}$ mark out of $N$ as an active mark shown in Eq. 1 to capture global audio semantic information.

Building upon the semantic-enhanced audio features, we carry out self-attention $\text{SA}(\cdot)$ with the residual of initialized learnable queries $q$ and obtain the semantic-aware queries $q'$ as

$$q' = \text{SA}(F'_A, F'_A, F'_A) + q = \text{softmax}\left(F'_A F'^{T}_A\right) F'_A + q, \tag{4}$$

where $q' \in \mathbb{R}^{t \times N \times d_A}$ is the semantic-aware queries. Here, to enhance conciseness, the equations in this paper for attention have excluded projection and scaling factors. The semantic-aware queries obtained are utilized as the learnable query inputs for the transformer decoder layers. Similarly, the queries in the same position corresponding to latent semantic active marks are *active queries* $q'_a$, while those corresponding to dormant marks are *dormant queries* $q'_d$, where $\{q'\} = \{q'_a\} \cup \{q'_d\}$. Moreover, the adequate number of active queries $q'_a$ per segment is ensured in multiple ways and please refer to the appendix for more details.

## 4.3 Enhancing Interaction within Active Queries

After introducing active and dormant queries, we observed that regions proposed by dormant queries are sometimes unnecessarily optimized and backpropagated. It is clearly detrimental since dormant queries do not contain adequate audio semantic information. The widely used transformer decoder layer in per-mask segmentation consists of three modules to process learnable query features $q \in \mathbb{R}^{N \times t \times d_A}$ in the following order: a cross-attention $\text{CA}(\cdot)$, a self-attention module $\text{SA}(\cdot)$, and a feed-forward network $FFN(\cdot)$. For the original transformer decoder, the interactions occur between all learnable query features, and the same query can propose regions of different semantics. Original cross-attention and self-attention in $l^{th}$ layer can be denoted as

$$q^{ca}_{(l)} = \text{CA}(q_{(l-1)}, F_{V_{(l)}}, F_{V_{(l)}}) + q_{(l-1)}$$
$$= \text{softmax}(\mathbf{m}_{(l-1)} + q_{(l-1)} F_{V_{(l)}}) F_{V_{(l)}} + q_{(l-1)}, \tag{5}$$
$$q^{sa}_{(l)} = \text{SA}(q^{ca}_{(l)}, q^{ca}_{(l)}, q^{ca}_{(l)}) + q^{ca}_{(l)}$$
$$= \text{softmax}(q^{ca}_{(l)} q^{ca \, T}_{(l)}) + q^{ca}_{(l)}, \tag{6}$$

where $l$ is the layer index, $q_{(l)}$ refers to the query features corresponding to the learnable queries $q$. $q^{ca}_{(l)}$ and $q^{sa}_{(l)}$ is the temporary

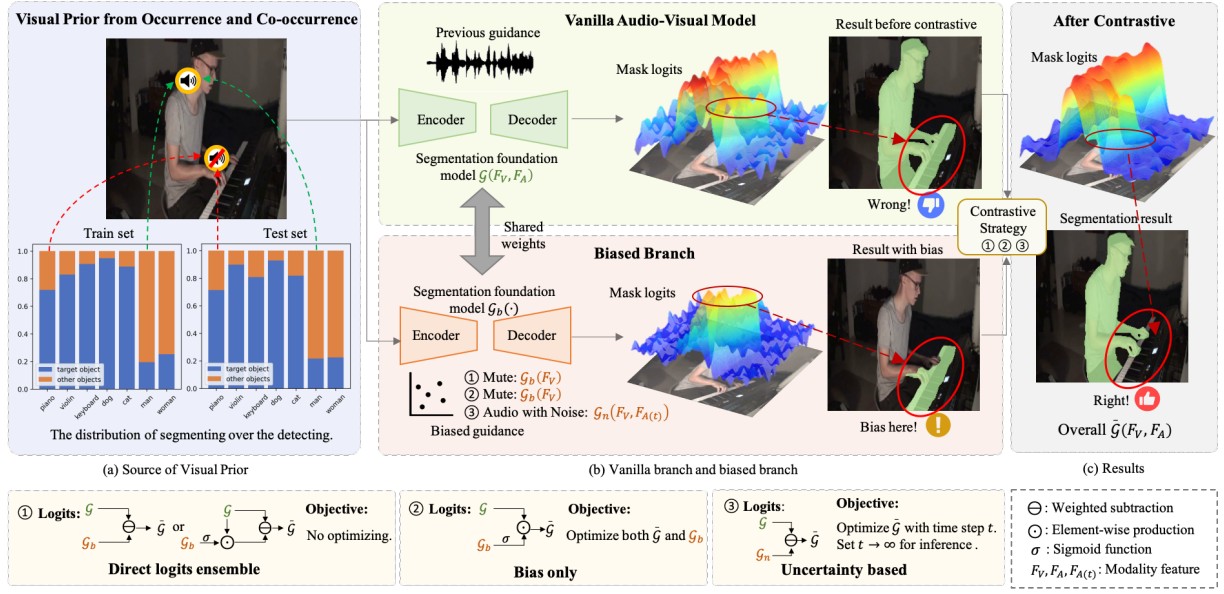

Figure 4: Illustration of visual prior. (a) The model always tends to learn statistically plausible results, rather than achieve the harder desired grounding behavior. Note: the bar chart uses blue to represent the proportion of the object being present in the image and emitting sound, while the orange color represents the proportion of the object being present but not emitting sound. As an example, since the piano generally appears with a high sounding probability in the training data, the model will segment once sees the piano. (b) To deal with the visual prior, we introduce debias strategies through the idea of contrasting the audio-visual model with the biased branch. (c) The ideal result is the mask without visual prior.

state after the cross-attention CA($\cdot$) and after self-attention SA($\cdot$) respectively. Note, $F_{V_{(l)}}$ are the image features under transformation, $\mathbf{m}_{(l)}$ is the masked attention introduced by Chen *et al.* [7].

However, we argue that audio interactions should only occur between active queries related to perceived latent semantics. This ensures that a specific set of active queries should only propose the region of the same semantics.

$$q_a^{ca}{}_{(l)} = \mathrm{CA}(q'_{a(l-1)}, F_{V_{(l)}}, F_{V_{(l)}}) + q'_{a(l-1)}, \qquad (7)$$

$$q_a^{sa}{}_{(l)} = \mathrm{SA}(q_a^{ca}{}_{(l)}, q_a^{ca}{}_{(l)}, q_a^{ca}{}_{(l)}) + q_a^{ca}{}_{(l)}, \qquad (8)$$

where only active query features $q'_{a(l-1)}$ are updated to $q'_{a(l)}$ by $FFN(q_a^{sa}{}_{(l)})$, and dormant query features $q'_{d(l)}$, are masked during backpropagation. Finally, based on the $q'_{a(l)}$ in each layer, the mask logits $\in \mathbb{R}^{T \times N \times H \times W}$ and class logits $\in \mathbb{R}^{T \times N \times \mathcal{K}}$ are obtained, where $\mathcal{K}$ is the number classes of semantics ($\mathcal{K}$=1 for binary).

Following each transformer decoder layer, query matching aims to determine which of the predicted regions fits the referred objects. Therefore, we match the predicted regions from the active queries and the ground truth regions by minimizing the matching cost

$$(\mathcal{M}, \mathcal{M}_{gt}) = \underset{\mathcal{M} \in \mathcal{M}_{q'_a}}{\arg\min} C_{\mathrm{match}}\left(\mathcal{M}_{q'_a}, \mathcal{M}_{gt}\right), \qquad (9)$$

where $\mathcal{M}_{q'_a}$ are the regions proposed by $q'_a$, and $\mathcal{M}_{gt}$ is the ground truth of original image. $\mathcal{M}_{q'_a}$ is calculated by multiplying mask logits and class logits, so the cost $C_{\mathrm{match}}$ evaluates the predicted region in both mask and semantic level. In practice, we separately compute the costs of mask and class logits and jointly optimize the summation as $C_{\mathrm{match}}$. Then, the losses $\mathcal{L}$ are calculated in every

intermediate transformer decoder layer, and only the one in the final layer is given a higher weight and used for inference.

Based on the method above, interactions related to active queries are retained and enhanced, while interactions of dormant queries are suppressed with a reduction of the computational workload. Although this method slightly reduces the number of proposed valid regions $\mathcal{M}_{q'_a}$, this trade-off is proven advantageous for better cooperation between audio semantics in further experiments.

## 4.4 Contrastive Debias Strategy

To recognize bias and then avoid the biased region appearing along with the referred region, it is a natural idea to introduce a new biased branch and conduct joint training to the current AVS model without modifying their structures. We first briefly bring up two simple and direct methods and carry out analysis and experiment for comparison. Subsequently, a well-designed uncertainty-based strategy is proposed with competitive performance and versatility. To maintain brevity, operations on *logits* in Sec. 4.4 are applied to both mask logits and class logits separately.

To begin with, the simplest approach to reorganize the distribution is the **direct logits ensemble strategy** of separately trained biased model $\mathcal{G}_b(F_V) \rightarrow \mathcal{M}_{gt}$ and vanilla AVS model $\mathcal{G}(F_V, F_A) \rightarrow \mathcal{M}_{gt}$, and subsequently merge their output logits using simple operations such as weighted subtraction and multiplication. As the second strategy, the **bias-only strategy** requires a new objective of training by involving a bias-only branch with the mapping of $\mathcal{G}_b(F_V) \rightarrow \mathcal{M}_{gt}$ and jointly optimize the overall model with a bias-only branch. To obtain the debias logits, we simply

compute an element-wise product $\odot$ of the logits of $\mathcal{G}$ and $\mathcal{G}_b$

$$logits(\bar{\mathcal{G}}(F_V, F_A)) = logits(\mathcal{G}(F_V, F_A)) \odot \sigma(logits(\mathcal{G}_b(F_V))), \quad (10)$$

where $\bar{\mathcal{G}}$ is the debias projection of the triplets $\mathcal{D}$ and $\sigma$ is the sigmoid function. The final optimization goal can be calculated from the logits as

$$\mathcal{L}_{\text{bias}}(\mathcal{G}, F_V, F_A, \mathcal{G}_b) = \mathcal{L}(\bar{\mathcal{G}}, F_V, F_A) + \mathcal{L}(\mathcal{G}_b, F_V), \quad (11)$$

Based on the updated logits, the region of the man in Fig. 4 will be given fewer logits and yield a greater loss than the original.

Although plausibly reorganizing the logits distribution, these two methods demonstrate fluctuations in experiments and do not consistently improve performance in popular methods. These issues can be attributed to the hard and direct introduction of the biased branch, which forces the network to learn from the large distribution gap, resulting in fluctuations in gradient direction and magnitude. To address this, we propose a soft and gradual strategy by introducing audio uncertainty through Gaussian noise.

Our proposed **uncertainty-based strategy** provides a gradual and soft way to estimate the biased output distribution. It does not require multitask-like loss but involves contrast $\mathcal{G}_n$ output distributions derived from image and distorted audio inputs. We first follow the idea of the forward diffusion process $r(F_{A(T)} \mid F_{A(0)})$ in image generation [16] to construct the distorted audio

$$r\left(F_{A(\tau)} \mid F_{A(\tau-1)}\right) = \mathcal{N}\left(F_{A(\tau)}; \sqrt{1-\beta}F_{A(\tau-1)}, \beta\mathbf{I}\right),$$
$$r\left(F_{A(T)} \mid F_{A(0)}\right) = \prod_{\tau=1}^{T} r\left(F_{A(\tau)} \mid F_{A(\tau-1)}\right), \quad (12)$$

where $F_{A(0)}$ denotes the original audio features without any noise and $I$ refers to an identity matrix. We incrementally add a small amount of Gaussian noise for $T$ steps. As step $\tau$ goes larger, the amount of noise added in each step is controlled by $\beta$. Finally, if $T \rightarrow \infty$, $F_{A(T)}$ will be completely distorted. Then, the final logits are updated as

$$logits(\bar{\mathcal{G}}(F_V, F_A)) = (\alpha + 1)logits(\mathcal{G}(F_V, F_A))$$
$$- \alpha logits(\mathcal{G}_n(F_V, F_{A(T)})), \quad (13)$$

where larger $\alpha$ values indicate a stronger amplification of differences between the two distributions ($\alpha = 0$ reduces to regular logits). During inference, $T \rightarrow \infty$, so $F_{A(T)}$ only contains Gaussian noise for the biased branch. Based on such logits, the overall loss

$$\mathcal{L}_{\text{uncertainty}} = \mathcal{L}(\bar{\mathcal{G}}, F_V, F_A), \quad (14)$$

is calculated by updated logits. Finally, the loss and mask result will be calculated from the debias $logits(\bar{\mathcal{G}}(F_V, F_A))$. In summary, replacing the hard bias-only branch with a gradually distorted audio branch provides a soft and learnable transition for the model to reorganize the distribution caused by visual prior.

## 4.5 Loss and Inference

Considering both masks and semantics, the overall loss $\mathcal{L}$ mentioned in Sec.4.4 is obtained within every intermediate transformer decoder layer by mask loss $\mathcal{L}_{mask}$ [24] and class loss $\mathcal{L}_{cls}$ based

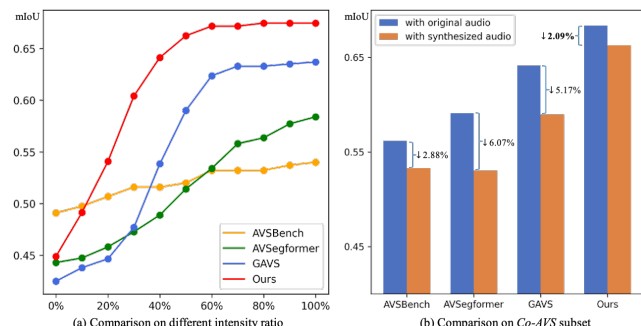

(a) Comparison on different intensity ratio

(b) Comparison on Co-AVS subset

**Figure 5: (a) The performance comparison of different methods on V1M under certain intensity conditions. Our method brings more sensitivity to low-intensity scenarios. (b) The performance comparison of different methods on our *Co-AVS* subsets. Our method has advantages in both performance and robustness.**

on the updated mask logits and class logits by debias strategy as

$$\mathcal{L} = \mathcal{L}_{mask} + \lambda_{cls} \cdot \mathcal{L}_{cls}, \quad (15)$$
$$\mathcal{L}_{mask} = \lambda_{focal} \cdot \mathcal{L}_{focal} + \lambda_{dice} \cdot \mathcal{L}_{dice}, \quad (16)$$

where $\mathcal{L}_{mask}$ is implemented by a combination of the dice loss $\mathcal{L}_{dice}$ [29] and the focal loss $\mathcal{L}_{focal}$ functions, and $\mathcal{L}_{cls}$ is implemented by the cross-entropy loss. After adopting the active queries, both losses $\mathcal{L}_{mask}, \mathcal{L}_{cls}$ are calculated and backpropagated within the matched pairs of masks $\mathcal{M}$ from active queries and ground truth $\mathcal{M}_{gt}$. During inference, the overall logits $\in \mathbb{R}^{(\mathcal{K}+1)\times H \times W}$ are obtained by the multiplication of mask logits and class logits in the last layer.

To address the shortcut effect resulting from easily learned regions proposed by specific queries, we drop the relatively well-learned pairs $(\mathcal{M}, \mathcal{M}_{gt})$ under the probability of $p$ in the calculation of $\mathcal{L}$ and empirical results show slight improvements.

## 5 EXPERIMENTS

### 5.1 Implementation Details

**Dataset.** Our proposed method is evaluated on the AVS Benchmarks [46], which contains three subsets. Firstly, the single-source subset (V1S) contains 4932 videos over 23 categories. In this subset, only the first sampled frame is annotated. Secondly, the multi-source subset (V1M) contains 424 videos that include two or more sound sources and all sounding objects are visible in the frames. Finally, the AVSBench-semantic (AVSS) subset, an extension of V1S and V1M, contains 12,356 videos of 10s, differing from the 5s videos in V1S and V1M. Additionally, to validate the cooperation performance in complex audio scenarios, a small multi-source test set of AVS called *Co-AVS* is created, containing 450 videos of 5s. Original audio clips are replaced with synthesized audio clips of 6 different levels intensities of each semantics, like the red block in Fig. 2.

**Setting.** We conduct training and evaluation using the VGGish backbone pretrained on Youtube-8M [1] and Swin-base Transformer backbone pretrained on semantic-ADE20K [44] by Mask2Former [7]. The parameters $\lambda_{focal}, \lambda_{dice}, \lambda_{cls}$ in the loss are set to 5, 5, 2, 1. The AdamW optimizer is adopted with a learning rate of 1e-4 for the visual encoder adapters and 1e-3 for other learnable parameters,

**Table 1: Comparison of performance on all three subsets of AVS-Benchmarks. The bold score is utilized to represent optimal results. "-" refers to the methods that do not support the corresponding subsets. Except for specific references, the "strategy" mentioned in the comparison refers to the uncertainty-based strategy that achieves the best outcomes.**

| Method | Audio-backbone | Visual-backbone | V1S | | V1M | | AVSS | |
|---|---|---|---|---|---|---|---|---|
| | | | mIoU(%) | F-score | mIoU(%) | F-score | mIoU(%) | F-score |
| AVSBench [46] | VGGish | PVT-v2 | 78.70 | 0.879 | 54.00 | 0.645 | 29.77 | 0.352 |
| AVSC [26] | VGGish | PVT-v2 | 81.29 | 0.886 | 59.50 | 0.657 | - | - |
| AVS-BG [14] | VGGish | PVT-v2 | 81.71 | 0.904 | 55.10 | 0.668 | - | - |
| AQFormer [18] | VGGish | PVT-v2 | 81.60 | 0.894 | 61.10 | 0.721 | - | - |
| AuTR [28] | VGGish | Swin-Base | 80.40 | 0.891 | 56.20 | 0.672 | - | - |
| BAVS [27] | BEATs | Swin-Base | 82.68 | 0.898 | 59.63 | 0.659 | - | - |
| SAMA [29] | VGGish | ViT-Huge | 81.53 | 0.886 | 63.14 | 0.691 | - | - |
| †CATR [22] | VGGish | PVT-v2 | 81.40 | 0.896 | 59.00 | 0.700 | 36.66 | 0.420 |
| AV-SAM [30] | ResNet18 | ViT-Base | 40.47 | 0.566 | - | - | - | - |
| AVSBG [14] | VGGish | PVT-v2 | 81.71 | 0.904 | 55.10 | 0.668 | - | - |
| GAVS [38] | VGGish | ViT-Base | 80.06 | 0.902 | 63.70 | 0.774 | - | - |
| AVSegFormer [10] | VGGish | PVT-v2 | 82.06 | 0.899 | 58.36 | 0.693 | 32.80 | 0.385 |
| MUTR [41] | VGGish | Video-Swin-Base | 81.60 | 0.897 | 64.00 | 0.735 | - | - |
| Ours (*w/o.* strategy) | VGGish | Swin-base | 82.92 | 0.928 | 66.12 | 0.792 | 41.93 | 0.476 |
| Ours (*w/.* strategy) | VGGish | Swin-base | **83.31** | **0.930** | **67.22** | **0.808** | **44.42** | **0.499** |

† CATR: To make a fair comparison, the results of CATR here are without supplemented annotation of the training set.

**Table 2: Ablation study of the methods and strategies proposed in this study on V1M.**

| Ablation | mIoU(%) | F-score |
|---|---|---|
| Ours | **67.22** | **0.808** |
| *w/o.* Contrastive debias strategy | 66.12 | 0.792 |
| *w/o.* Dropping dominant queries | 65.95 | 0.790 |
| *w/o.* Interaction enhancement | 62.57 | 0.744 |
| *w/o.* Active query of $0^{th}$ | 62.36 | 0.743 |
| *w/o.* Active queries | 62.16 | 0.734 |

with 60 training epochs. $N$ and $k$ in Fig. 3 is set to 100 and 9. Drop possibility $p$ in Sec. 4.5 and $\alpha$ in Eq. 13 are set to 0.4 and 1 without elaborate searching. $T$ related to the uncertainty level in Eq.12 are set to 1000.

**Metrics.** To conduct a comprehensive evaluation of our model, we carry out tests using mean Intersection over Union (mIoU) and F-score as the performance metrics [10, 38, 46].

## 5.2 Results Comparison

When compared to methods that do not incorporate the designed debias method, our method, which leverages methods for both biases, demonstrates strong competitiveness. As illustrated in Tab. 1, our method achieves comparable results across all V1S, V1M, and AVSS subsets, showcasing performance improvements of up to 0.63%, 3.22%, 7.76% in mIoU, respectively. Even when considering slight variations in the backbones used by different methods, our approach consistently outperforms others. The limited improvement observed in V1S can primarily be attributed to the constrained manifestation of biases stemming from the singular and simplistic nature of the data. However, we still believe that this does not undermine our ability to demonstrate the effectiveness of our bias handling techniques.

Furthermore, both quantitative performance in Fig. 5 and qualitative analysis in Fig. 6 indicate that our model successfully alleviates

**Table 3: The analysis of projection strategy to obtain semantic-aware queries by class or cluster. The ablation of $topK$ can also be regarded as the ablation of a fixed number of active queries.**

| method | filtering | mIoU(%) | F-score |
|---|---|---|---|
| Classification | $\theta = 0.8$ | 64.55 | 0.757 |
| | $\theta = 0.6$ | 66.36 | 0.783 |
| | $\theta = 0.4$ | **67.22** | **0.808** |
| | $\theta = 0.2$ | 64.52 | 0.740 |
| | $topK = 1$ | 61.79 | 0.695 |
| | $topK = 3$ | 64.49 | 0.742 |
| | $topK = 5$ | 63.81 | 0.719 |
| Clustering | Kmeans | 64.18 | 0.746 |
| | GMMs | 64.57 | 0.742 |
| None | None | 63.59 | 0.702 |

both biases mentioned before and our predicted masks exhibit superior quality. Furthermore, due to the increased complexity of synthesized audio scenarios in *Co-AVS* subset, achieving a lower performance drop on synthesized audio necessitates better semantic collaboration. As shown in Fig.5, our demonstrates stronger robustness on complex audio scenarios.

## 5.3 Ablation Study

**Overall ablation.** We conduct an ablation experiment on the module and the strategy designed for biases. In this experiment, we progressively remove the modifications of each module in the set of V1M. Through this process, we aimed to assess the impact of these modifications on the segmentation results. As demonstrated in Tab. 2, the experiments indicate that the methods we designed for audio priming bias and visual prior respectively achieved an increase of 3.96% and 1.10% in mIoU. Different modules have all achieved an increase in both mIoU and F-score, reaching the design objectives. The ablation study further indicates that the inclusion of active queries alone does not directly result in a significant increase in

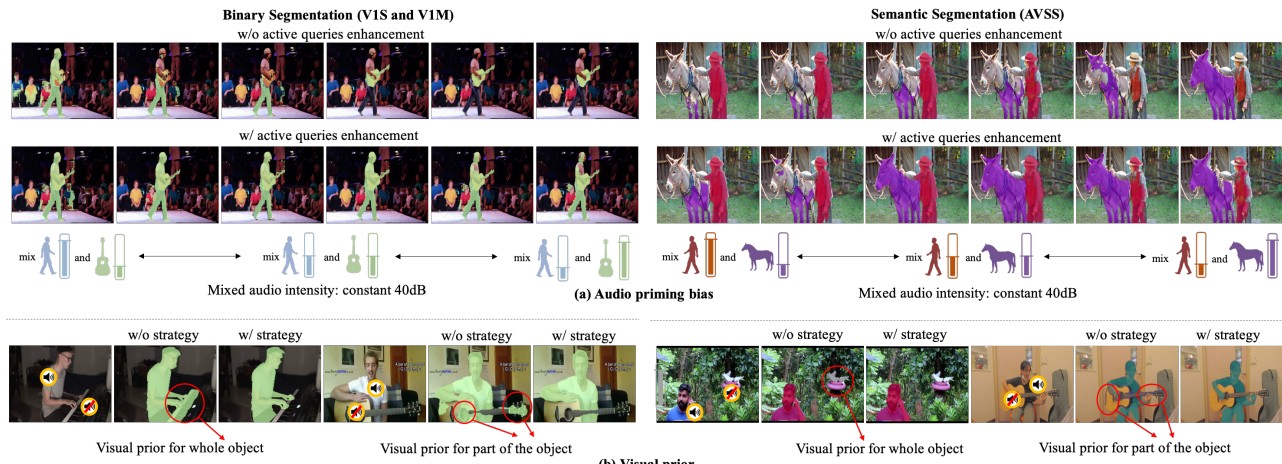

Figure 6: Our model successfully alleviates both biases and our predicted masks exhibit superior quality than before.

Table 4: Ablation analysis and the versatility experiments of image-biased strategy on V1M. "↑" signifies a positive effect achieved by employing the contrastive debiased strategy compared to the vanilla method, while "↓" indicates a negative effect.

| Method | Backbone | Vanilla | | Logits ensemble | | Bias only | | Uncertainty based | |
|---|---|---|---|---|---|---|---|---|---|
| | | mIoU (%) | Fscore | mIoU (%) | Fscore | mIoU (%) | Fscore | mIoU (%) | Fscore |
| AVSbench [46] | PVT-v2 | 54.00 | 0.645 | 54.12↑ | 0.659↑ | 53.43↓ | 0.640↓ | 54.81↑ | 0.682↑ |
| AVSegformer [10] | PVT-v2 | 58.36 | 0.693 | 56.85↓ | 0.662↓ | 58.86↑ | 0.687↓ | 59.63↑ | 0.718↑ |
| GAVS [38] | Swin-base | 63.70 | 0.766 | 51.21↓ | 0.738↓ | 64.11↑ | 0.788↑ | 64.80↑ | 0.784↑ |
| Ours | Swin-base | 66.12 | 0.792 | 64.42↓ | 0.795↑ | 66.50↑ | 0.797↑ | **67.22↑** | **0.808↑** |

performance. It necessitates the implementation of enhancement of interaction, ultimately leading to a combined improvement of 3.76%. This finding emphasizes the importance of enhancing interaction in conjunction with active queries, as they aid the transformer encoder in establishing audio-visual grounding.

**Ablation on projection of Equation 2.** The objective of projection is to map each audio clip to several classes or a cluster with latent semantic information. Therefore, we naturally conducted experiments using both supervised and unsupervised approaches. In the supervised experiments, we employed Beats [6] for supervised multi-class classification and used threshold $\theta$ or top$K$ strategies to roughly filter the samples with abstract semantic information in Tab. 3. The results showed that the different settings in the multi-classification do give an improvement of mIoU on V1M up to 67.22%. In the unsupervised experiments, we utilized the K-means and Gaussian Mixture Model (GMM) with Expectation Maximization for clustering, with the same number of clusters. Moreover, an adequate number of active queries per sample is ensured here for fairness. The results demonstrated that, regardless of supervised or unsupervised approaches, introducing latent semantic information along with active queries consistently improved performance, with a maximum improvement of 3.63% in Tab. 3.

**Mask of single active queries.** Besides the quantitive results in Tab. 2, it is qualitatively found that the masks generated by each active query are consistent with their respective semantics, which further validates the effectiveness of the semantic-aware active queries in audio perception. Detailed qualitative results of cases are provided in the appendix.

## 5.4 Versatility of Contrastive Debias Strategies

As general strategies, all three contrastive debias strategies are able to reorganize the distribution of logits. To validate the versatility, we not only applied the debias strategies in our framework but also explored its application in other popular models, as shown in Tab. 4. The performance of the logits ensemble and bias-only strategy fluctuates, achieving performance improvements only in certain methods. When considering all four methods, the uncertainty-based approach emerges as a superior solution due to its general improvement. Within our framework, it attained mIoU of 67.22% and exhibited commendable training stability.

## 6 CONCLUSION

This study presents a novel and systematic exploration of the bias issues existing in the previous AVS model. By defining and analyzing these biases as "audio priming bias" and "visual prior", we propose targeted solutions for each bias. For audio priming bias, we introduce semantic-aware active queries that enhance the audio sensitivity of intensity and semantics with a designed perception module. Then further interaction of active queries is forced to collaborate on semantic understanding. For the visual prior, we employ a contrastive debias strategy that improves the overall performance of the model without modifying its structure. The meticulously designed methods and comprehensive experiment across all three subsets, demonstrate the superiority and versatility of our approach. In summary, our work introduces a novel perspective within the prevailing AVS network, opening up new possibilities for the future development of cross-model guided segmentation.

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
