# OpenReview forum: "Unveiling and Mitigating Bias in Audio Visual Segmentation"
_acmmm.org/ACMMM/2024/Conference — MM2024 Oral_

### Official Review · Reviewer_Q3Hw · 2024-05-23

**Rating:** 4
**Confidence:** 2

**Summary:**

This study identifies and addresses "audio priming bias" and "visual prior" in previous Audio-Visual Segmentation (AVS) models. To mitigate these biases, the authors introduce semantic-aware active queries for enhanced audio sensitivity and a contrastive debias strategy to improve visual performance without altering the model structure. Comprehensive experiments demonstrate the effectiveness of these methods, offering new insights for future development in cross-modal guided segmentation.

**Strengths:**

The strength of this paper lies in its novel definition of "audio priming bias" and "visual prior" as the sources of bias in Audio-Visual Segmentation (AVS). Additionally, the proposed solutions, including semantic-aware active queries and a contrastive debias strategy, are well-structured and effectively address these biases, enhancing the overall performance and robustness of the AVS model.

**Limitations:**

While the paper presents a novel approach and effective solutions, there are some areas that could benefit from further explanation:

[1] Line 387-388: “These queries are simply inherited by repeatedly adding audio features.” Can you explain more about how the queries are generated?

[2] Could you elaborate on the relationship between the number of learnable queries and the visual size of HxW?

[3] Can you provide more details about the dimensions related to Equations (1) and (2)? Specifically, does Equation (1) imply there are txN elements? If so, are the elements defined in Equation (2) also defined for txN elements?

[4] In Equation (1), what is the intuition behind designating the first element as 1 for capturing global audio semantic information?

[5] Line 507: Could you clarify the dimensions of mask logits and class logits and explain why N is needed? Additionally, Could you describe the dimensions from the input to the logits step by step? Plus, it would be more consistent to use 't' instead of 'T' in Line 507.

[6] Line 389: Figure reference should be corrected from Fig. 2 to Fig. 3.

[7] Equation 9: The notation 𝑀_{𝑞’_𝑎} should be corrected to 𝑀.

**Suitability:**

3

---

### Official Review · Reviewer_eNsj · 2024-05-26

**Rating:** 5
**Confidence:** 3

**Summary:**

This paper addresses the critical issue of biases in Audio-Visual Segmentation (AVS) models, specifically focusing on two types of biases: "audio priming bias" and "visual prior." The authors introduce novel methods to mitigate these biases, including semantic-aware active queries for enhancing audio sensitivity and a contrastive debias strategy for addressing visual prior. The paper presents extensive experimental evaluations, demonstrating the effectiveness of the proposed methods across multiple AVS benchmarks, achieving competitive performance and improving the quality of sounding objects' masks.

**Strengths:**

1. Novelty and Theoretical Approach: The paper introduces innovative concepts such as "audio priming bias" and "visual prior," which are well-defined and systematically analyzed. The proposed solutions, including semantic-aware active queries and contrastive debias strategies, represent significant advancements in the field of AVS.

2. Technical Correctness: The methods are technically sound, leveraging advanced techniques like transformer decoders, semantic-aware queries, and contrastive learning. The detailed mathematical formulations and logical flow of the proposed methods indicate a high level of technical rigor.

3. Comprehensive Evaluation: The authors conduct extensive experiments on multiple AVS benchmarks, including V1S, V1M, and AVSS subsets, as well as a newly created Co-AVS subset. The results are presented in a clear and detailed manner, showing significant improvements in performance metrics such as mIoU and F-score.

4. Clarity and Organization: The paper is well-structured and clearly written, with each section logically building upon the previous one. The figures and tables are effectively used to illustrate the key points and results.

5. Applications and Impact: The proposed methods have broad implications for improving AVS models, which are crucial for various applications in multimedia processing, autonomous driving, and embodied intelligence. The systematic approach to addressing biases enhances the robustness and accuracy of AVS models in real-world scenarios.

**Limitations:**

1. Evaluation Scope: While the paper presents a comprehensive evaluation on multiple benchmarks, it would benefit from additional comparisons with more recent and diverse AVS methods to further validate the generalizability of the proposed approaches.

2. Complexity and Implementation: The proposed methods involve sophisticated techniques that may be challenging to implement and require substantial computational resources. A discussion on the computational complexity and potential optimizations would be valuable.

3. Synthetic Data: The use of synthetic data for certain experiments, while useful for controlled studies, may not fully capture the complexity of real-world audio-visual scenarios. More experiments with real-world data would strengthen the conclusions.

**Suitability:**

3

---

### Official Review · Reviewer_7dXz · 2024-05-26

**Rating:** 4
**Confidence:** 3

**Summary:**

The paper presents a new exploration on bias issues prevalent in the exiting audio-visual models. To this end, the authors discuss two critical aspects by formulating audio priming bias and visual prior bias. The problem is very relevant to the community and is underexplored. The paper presents an intuitive approach to tackle these challenges and experimental results show that the presented method achieves considerable improvements over the prior baselines.

**Strengths:**

1. The paper presents two very important anomalies commonly observed in audio-visual space : "audio priming bias" and "visual prior" and explores possible ways to mitigate these issues in existing approaches. These problems are pressing yet underexplored in the community this exploration can serve as the ground work for future investigations in this direction.
2. The authors present intuitive ways to tackle these challenges and through experimental analysis the paper demonstrates the efficiency of the proposed solutions.
3. The paper is generally well written, easy to follow, although I would recommend the authors to consider re-writing some parts of the papers (especially introduction and problem motivation).

**Limitations:**

1. Although the problem is very relevant to the community and needs attention, the paper in patches lack motivating the problem statement clearly and can do a better job there. I would recommend the authors to consider laying a strong foundation by adding more practical and real world examples and connecting that with the failure models in existing works and how their proposed solution is better equipped at tackling these issues.
2. The paper doesn't discuss some important literatures in this space:
[a] Self-supervised Audio-Visual Co-segmentation - Rouditchenko et al.
[b] Weakly-Supervised Audio-Visual Segmentation - Mo et al.
[c] Listen to the pixels - Chowdhury et al.
These works were initial attempts in the audio-visual segmentation area.
3. I might be missing this, but why does the authors assign each audio segment into 8 classes? Is there a theoretical or empirical baking behind this?
4. Have the authors tried training their model with hard negatives? This can serve as a baseline for the visual prior task at least. This analysis can clearly underline the drawbacks in existing approaches and provide us with an estimate of the performance gain
5. The contrastive debias strategy is interesting, I wonder what physical aspects aspects do their approach consider while learning this debiasing? How much of a role does the datasets play and what are some good ways the authors think can be applied reuse the existing datasets with limited data distillation?
6. In table 4 there are quite a few cases where  employing the contrastive debiased strategy results in performance drop. Can the authors explain this behaviour or at least add some intuition what might be the reasons behind this observation?
7. The could have considered adding some physical examples through a video in the supplementary for better visualization.
8. The experimental section looks a bit weak. More extensive experiments on the choice of encoder, training strategy (effect of number of input tokens), comparisons in setting with >2 sounding objects are missing

Minor:
1. grammatical mistakes 'there is an urgent.. ' line 88
2. Can the authors explain in brief their reasoning behind employing swin backbone?

**Suitability:**

3

---

### Official Review · Reviewer_hDtv · 2024-05-30

**Rating:** 3
**Confidence:** 3

**Summary:**

This paper analyzes the issue of bias in audio-visual segmentation models which may lead to inaccurate segmentation results. The authors categorize two main types of biases, "audio priming bias" and "visual prior". For "audio priming bias", they propose semantic-aware active queries to enhance audio sensitivity and customized interaction mechanisms in the transformer decoder for better audio-semantic collaboration. For  "visual prior", they explore contrastive debias strategies that optimize the model without structural changes. They conduct experiments on AVS benchmarks and achieve competitive performance compared with prior methods.

**Strengths:**

- **Novelty**: The paper introduces an approach to identifying and mitigating biases in the task of audio-visual segmentation, which has not been explored in prior works.
- **Technical Correctness**: The technical solutions proposed in the paper appear to be correct and well-implemented. The authors have detailed the methodology for enhancing audio sensitivity and addressing visual prior biases with clarity. The use of contrastive debias strategies, especially the uncertainty-based approach, is technically innovative.

**Limitations:**

- **Motivation**: The two types of bias come from the VQA task. The authors replace "Visual priming bias" and "language prior" in the VQA task with "audio priming bias" and "visual prior" in the AVS task. The issues observed from the AVS task itself are more appreciated.
- **Datasets**: Do V1S and V1M subsets represent S4 and MS3 subsets in the AVS benchmarks? If not, why not compare with previous methods on these two subsets?
- **Ablation studies**: Table 2 did not list the result of the vanilla model without all proposed modules and the comparison with it.

**Suitability:**

3

---

### Meta-Review · Area_Chair_Gn3J · 2024-06-30

**Recommendation:** Accept (Oral)
**Confidence:** 4

**Metareview:**

The paper is well written. The reviewers acknowledge the novelty and technical correctness of this work and are satisfied with the response. I am delighted to recommend the acceptance of this paper.